# UniAIDet: A Unified and Universal Benchmark for AI-Generated Image Content Detection and Localization

## Abstract

With the rapid proliferation of image generative models, the authenticity of digital images has become a significant concern. While existing studies have proposed various methods for detecting AI-generated content, current benchmarks are limited in their coverage of diverse generative models and image categories, often overlooking end-to-end image editing and artistic images. To address these limitations, we introduce UniAIDet, a unified and comprehensive benchmark that includes both photographic and artistic images. UniAIDet covers a wide range of generative models, including text-to-image, image-to-image, image inpainting, image editing, and deepfake models. Using UniAIDet, we conduct a comprehensive evaluation of various detection methods and answer three key research questions regarding generalization capability and the relation between detection and localization. Our benchmark and analysis provide a robust foundation for future research.[1]

## 1 Introduction

Image generative models have demonstrated impressive capabilities in synthesizing realistic images (Sauer et al., 2022; Betker et al., 2023; Wu et al., 2025a). However, this powerful capability raises significant concerns regarding image authenticity. Malicious use cases, such as deepfakes, involve passing off generated images as real to spread misinformation, or seeking improper profits by selling AI-generated artworks as human-created pieces.

Numerous studies have been conducted to address this problem. The majority of this research, however, concentrates solely on detecting entirely generated images. For instance, benchmarks like GenImage (Zhu et al., 2023) and AIGCDetect (Zhong et al., 2023), as well as methods such as (Chen et al., 2024a; Yan et al., 2024a; Tan et al., 2024b), are all primarily designed for binary classification and overlook fine-grained localization. Furthermore, they often neglect artistic content, which significantly limits their generalization capabilities. ImagiNet (Boychev & Cholakov, 2025) is a benchmark containing both photographic and artistic images, yet it still fails to cover partly generated images and support localization task.

Other studies have extended this task to include localization, a critical capability for identifying the generated parts within an image. Notable examples include benchmarks and methods such as FakeShield (Xu et al., 2024) and SIDA (Huang et al., 2025) for photos, and DeepFakeArt (Aboutalebi et al., 2023) for paintings. However, these works typically consider a very limited range of generative methods (often only one or two), which significantly restricts the generalization capability of their proposed benchmarks and methods. Furthermore, they universally neglect a crucial and increasingly popular category of generative models: instruction-guided image editing models, such as (Brooks et al., 2023).

With the aforementioned shortcomings in existing studies, there are several key problems lacking thorough discussion, including the relation between detection and localization and the generalization

---

[1]We release a subset of our benchmark in `https://anonymous.4open.science/r/UniAIDet-46D3` since the whole benchmark is too large to be anonymously shared. We will make the whole benchmark public upon acceptance.

Table 1: Examples of test images in our benchmark. T2I refers to images generated through a text-to-image model. I2I refers to images generated using image-to-image models. Edit refers to images generated using instruction guided image editing models. Inpaint refers to images generated using inpainting models. DeepFake refers to images generated using deepfake methods. For Edit, Inpaint, and DeepFake images, we label the generated parts in red.

| | Real | T2I | I2I | Edit | Inpaint | DeepFake |
|---|---|---|---|---|---|---|
| Photo | | | | | | |
| Art | | | | | | |

capability of existing detection and localization methods, especially on partly generated images and on different categories of images.

In this study, we introduce UniAIDet, a unified and universal benchmark for AI-generated image content detection and localization. Our benchmark encompasses a diverse range of image categories, including both photographic and artistic images (such as famous paintings and popular anime). It covers the outputs of a wide array of generative models, including text-to-image, image-to-image, image editing, image inpainting, and deepfake models. What's more, each partial generated image (e.g. inpainting, editing and deepfake) is meticulously provided with a mask that indicates the specific AI-generated regions for evaluating localization methods. Comprising a total of 80k real and generated images and covering 20 generative models, our benchmark represents the first large-scale and wide-coverage dataset for this task. Examples from our benchmark are presented in Table 1.

Using our benchmark, we conducted a comprehensive evaluation of various detection and localization methods, including those designed for detection-only and joint detection&localization models. Our results reveal that existing methods perform poorly on our benchmark, underscoring its value in exposing current limitations.

Furthermore, we address three key research questions, including the relation between detection and localization, the generalization capability on different generative models, and the generalization capability on different categories of images. Our findings suggest that a good detection performance generally leads to good localization performance, and generalization is still a challenging problem.

To summarize, our contributions are listed as follows:

- We propose the first large-scale, wide-coverage benchmark AI-generated image content detection and localization, covering most potential practical scenarios.
- We conduct a wide range of evaluation, revealing the shortcomings of previous AI-generated image detection and localization methods.
- We perform deep analysis and answer three important research questions regarding generalization and the relation between detection and localization, providing essential conclusions for future research.

## 2 RELATED WORKS

Numerous benchmarks have been proposed to evaluate AI-generated image detectors, including widely used datasets like UniversalFakeDetect (Ojha et al., 2023), GenImage (Zhu et al., 2023),

and AIGCDetect Zhong et al. (2023). SIDBench (Schinas & Papadopoulos, 2024) ensembles existing benchmarks to provide a larger benchmark. More recently, WildFake (Hong & Zhang, 2024), Chameleon (Yan et al., 2024a), and AIGIBench (Li et al., 2025b) were introduced to incorporate newer generative models. However, these benchmarks only support binary classification, lacking fine-grained localization capabilities and containing no artistic images, which significantly limits their application range.

Benchmarks designed for both AI-generated image detection and localization have also emerged. Recent studies like FakeShield (Xu et al., 2024) and SID-Set (Huang et al., 2025) focus on photos, while DeepFakeArt (Aboutalebi et al., 2023) is dedicated to artistic images. These benchmarks provide both generated images and corresponding masks that indicate the synthesized regions. However, a significant limitation is their narrow coverage of generative models. Specifically, they contain very few generative models, which may lead to biased evaluation results. Furthermore, they universally overlook a crucial and widely used type of generative model: instruction-guided image editing models. While LEGION (Kang et al., 2025) proposes another benchmark, its focus is on detecting artifacts rather than generated content, representing a slightly different research context.

Numerous detection methods have been proposed to address this problem. Early research efforts, such as those by (Wang et al., 2020) and (Liu et al., 2020), served as foundational works for detecting AI-generated images. Subsequent studies leveraged models like CLIP (Ojha et al., 2023) and its finetuned variants (Yan et al., 2024b; Tan et al., 2025; Keita et al., 2025), while other methods include DRCT (Chen et al., 2024a) and NPR (Tan et al., 2024a). Furthermore, many approaches have utilized frequency-based features to improve detection, including FreqNet (Tan et al., 2024b), AIDE (Yan et al., 2024a), and SAFE (Li et al., 2025a). There are also methods using reconstruction to train a detection model, such as DIRE (Wang et al., 2023) and FIRE (Chu et al., 2025). A significant limitation of these methods, however, is that they lack the ability to perform localization on partially generated images.

Some methods are designed to support both detection and localization, such as HiFi-Net (Guo et al., 2023). More recent approaches, including FakeShield (Xu et al., 2024) and SIDA (Huang et al., 2025), additionally incorporate reasoning capabilities. However, a significant limitation of these methods is their evaluation on a very restricted set of generative models and photo-only content, which makes their generalization capabilities dubious.

## 3 BENCHMARK CONSTRUCTION

### 3.1 TASK DEFINITION AND BENCHMARK OVERVIEW

The task of AI-generated image content detection and localization can be formalized as follows. Given an image $I \in \mathbb{R}^{C \times H \times W}$ with pixels $I_{i,j}$, a **real** image is defined as one where no pixel $I_{i,j}$ is produced by an AI generative model. Conversely, a **synthetic** image contains at least one pixel $I_{i,j}$ that is AI-generated. The **detection** task is a binary classification problem to determine whether an image is real or synthetic. The **localization** task aims to identify the set of pixel positions $\{(i,j)\}$ corresponding to the AI-generated regions.

We categorize practical AI generative models into two categories: **holistic synthesis models**, which produce a completely synthetic image, and **partial synthesis models**, which generate a partially modified image with some regions synthetic while others remain unchanged from a real image. Holistic synthesis models include text-to-image and image-to-image models (we do not discuss unconditional generative models, as they are not widely used in practice today). Partial synthesis models, on the other hand, include image inpainting, image editing, and deepfake methods.

It is important to note two key factors. First, unlike LEGION (Kang et al., 2025), our localization task directly focuses on whether a pixel was produced by an AI generative model, irrespective of its realism or the presence of artifacts. We argue that the core issue with generated images stems from the generative process itself, not from the lack of realism in the final output.

Secondly, we define purely human-created paintings and drawings as real images, as their creation process does not involve AI generative models. This definition aligns with the requirements of the community, where users often prefer works created by human artists over those generated by AI.

Table 2: Comparison between our proposed benchmark and existing benchmarks. ✓ refers to only containing some kind of partial synthesis model instead of all kinds of partial synthesis models.

| Benchmark | Content (Photo & Art) | Holistic Synthesis Model | Partial Synthesis Model | Localization | #Generative Models |
|---|---|---|---|---|---|
| DeepFakeArt | ✗ | ✓ | ✓ | ✓ | 2 |
| GenImage | ✗ | ✓ | ✗ | ✗ | 8 |
| ImagiNet | ✓ | ✓ | ✗ | ✗ | 7 |
| AIGCDetct | ✗ | ✓ | ✗ | ✗ | 16 |
| WildFake | ✗ | ✓ | ✗ | ✗ | 23 |
| Chalmeon | ✗ | ✓ | ✗ | ✗ | Unk |
| AIGIBench | ✗ | ✓ | ✓ | ✗ | 25 |
| FakeShield | ✗ | ✗ | ✓ | ✓ | 2 |
| SID-Set | ✗ | ✓ | ✓ | ✓ | 2 |
| UniAIDet(Ours) | ✓ | ✓ | ✓ | ✓ | 20 |

Table 2 provides a comprehensive comparison of our proposed benchmark with popular existing ones. Our benchmark stands out by offering a comprehensive evaluation for AI-generated image detection and localization methods, thanks to its wide coverage of image categories (photographic and artistic), diverse generative model categories, and a large number of included generative models. In contrast, other benchmarks exhibit significant limitations in one or more of these aspects, such as a lack of coverage on partial synthesis models.

## 3.2 REAL IMAGES COLLECTION

To construct a universal benchmark, we first collected real images from multiple resources. For photographic images, we sourced real images from MSCOCO (Lin et al., 2014) and NYTimes800k (Tran et al., 2020). NYTimes800k was specifically included to enhance the diversity of our benchmark. For artistic images, we sourced images from WikiArt (wik) [2] and Danbooru [3]. We specifically selected data created prior to 2023 to ensure the images were predominantly human-generated.

To ensure the quality of our benchmark, we applied an NSFW detector (Dosovitskiy et al., 2020; Falconsai, 2023) to the collected images to filter out potentially inappropriate content. These real images are collected from open-source datasets and used for research only, fully respecting their copyright.

## 3.3 SYNTHETIC IMAGES GENERATION

We further categorize generative models into five sub-categories, including text-to-image models and image-to-image models (which are holistic synthesis models), image inpainting models, image editing models, and deepfake methods (which are partial synthesis models). The detailed generation process for each category is presented as follows. Other details about our benchmark construction can be found in Appendix A.

**Text-to-Image Models**    For each text-to-image model, we first source several real images and generate captions using Gemma3-4B (Team, 2025). The resulting captions are then used as prompts for the text-to-image models to generate synthetic images. We utilize 8 open-source models, including Stable-Diffusion-1.5 (Rombach et al., 2022), Stable-Diffusion-XL (Podell et al., 2023), Pixart-Sigma-XL-ms (Chen et al., 2024b), FLUX.1-Dev (Labs, 2024), Stable-Diffusion-3 (Esser et al., 2024), Stable-Diffusion-3.5-Medium (Esser et al., 2024), Stable-Diffusion-3.5-Large-Turbo (Esser et al., 2024), and Qwen-Image (Wu et al., 2025a), along with one closed-source model, SeedDream (Gao et al., 2025).

---

[2] https://www.kaggle.com/datasets/ipythonx/wikiart-gangogh-creating-art-gan
[3] Data sourced from https://huggingface.com/datasets/animelover/danbooru2022, https://www.kaggle.com/datasets/mylesoneill/tagged-anime-illustrations

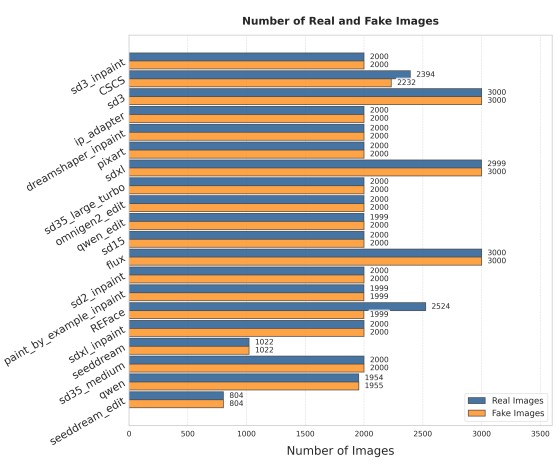 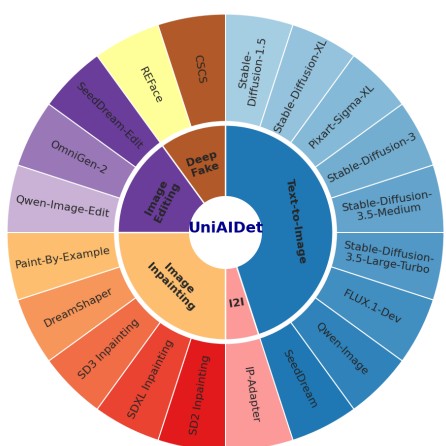

Figure 1: Data distribution of our benchmark.

Figure 2: Overview of generative models used in the benchmark.

**Image-to-Image Models** For each image-to-image model, we first source several real images and generate captions using Gemma3-4B (Team, 2025). The resulting captions, along with the original images, are then used as input to the image-to-image models to generate synthetic images. We utilize one open-source model: IP-Adapter (Ye et al., 2023).

**Image Inpainting Models** For each image inpainting model, we source several real images and generate masks using Segment-Anything-Model (Ravi et al., 2024). Each image is then fed into the image inpainting model along with a randomly selected mask to generate partially synthetic (inpainted) images. After generation, we clip the synthesized region (following the mask) from the generated image and paste it back into the original image. The combined image then serves as the final synthetic image. The mask is used as the ground truth of the generated region for evaluating localization methods. We utilize five open-source inpainting models: SD2 Inpaint (Rombach et al., 2022), SD3 Inpaint (Esser et al., 2024), SDXL Inpaint (Podell et al., 2023), DreamShaper-v8 (Lykon, 2023), and Paint-By-Example (Yang et al., 2023).

**Image Editing Models** For each image editing model, we source several real images and generate editing instructions using Gemma3-4B (Team, 2025). We do not restrict the range of editing prompts to improve the diversity of generated images. Our obtained editing prompts include add, remove, modify an object, background, and style. The resulting instructions, along with the original image, are then provided as input to the image editing models to generate edited images. After generating the edited images, we compare each edited image with its original counterpart to identify regions with notable changes as masks. We retain only the regions that are large enough, which are then pasted back into the original images to create the final synthetic images. The mask of these regions serve as the ground truth of the generated area for evaluating localization methods. We utilize two open-source image editing models: Qwen-Image-Edit (Wu et al., 2025a) and OmniGen-2 (Wu et al., 2025b), alongside one closed-source model, SeedDream-Edit (Gao et al., 2025).

**DeepFake Methods** We utilize the FFHQ (Karras et al., 2019) dataset as our source for facial images. We then perform face swapping using faces from FFHQ on several sourced real images to generate synthetic images. We use two open-source deepfake methods, REFace (Baliah et al., 2024) and CSCS (Huang et al., 2024), to generate these synthetic images. The mask produced by these methods is used as the ground truth of the generated regions.

### 3.4 BENCHMARK DETAILS

We present basic statistics of our benchmark in Figure 1 and an overview of the generative models included in Figure 2. More details can be found in Appendix A.

Table 3: Main results of different methods on two splits of our benchmark. AP is not calculated for non-threshold methods.

| Method | Photo | | | | | Art | | | | |
|---|---|---|---|---|---|---|---|---|---|---|
| | Acc | AP | f.Acc | r.Acc | mIoU | Acc | AP | f.Acc | r.Acc | mIoU |
| *Detection-Only* | | | | | | | | | | |
| CLIP | 66.12 | 70.15 | 56.26 | 75.96 | - | 62.37 | 65.66 | 59.11 | 65.79 | - |
| C2P-CLIP | 61.53 | 75.46 | 23.32 | 99.27 | - | 59.33 | 65.11 | 24.54 | 94.32 | - |
| DeeCLIP | 68.90 | 68.40 | 42.76 | 94.64 | - | 64.25 | 61.55 | 40.52 | 88.17 | - |
| DIRE | 50.64 | 49.21 | 1.54 | 98.38 | - | 49.98 | 52.90 | 1.43 | 98.55 | - |
| FIRE | 52.26 | 53.91 | **97.09** | 8.74 | - | 50.00 | 47.46 | **99.97** | 0.03 | - |
| Effort | 71.00 | 75.01 | 68.30 | 73.59 | - | 68.40 | 75.41 | 58.70 | 78.31 | - |
| DRCT | 75.13 | 84.47 | 51.84 | 98.02 | - | 71.41 | 77.69 | 59.41 | 83.63 | - |
| NPR | 72.51 | 78.72 | 47.51 | 97.04 | - | 71.97 | 80.52 | 46.34 | 97.87 | - |
| FreqNet | 67.73 | 72.83 | 62.27 | 73.15 | - | 68.06 | 74.66 | 52.48 | 83.86 | - |
| AIDE | 73.81 | 80.51 | 51.89 | 95.21 | - | 72.76 | 80.91 | 49.12 | 96.66 | - |
| SAFE | **79.89** | **88.44** | 59.83 | **99.51** | - | **79.15** | **85.89** | 59.96 | **98.62** | - |
| *Detection & Localization* | | | | | | | | | | |
| HiFi-Net | 60.69 | 68.58 | 22.73 | 97.45 | 3.79 | 62.30 | 69.25 | 28.69 | 96.00 | 6.55 |
| FakeShield | 68.53 | - | 82.05 | 55.43 | **17.55** | 60.41 | - | 88.62 | 32.27 | **14.24** |
| SIDA | 60.15 | - | 49.75 | 70.50 | 9.99 | 65.04 | - | 39.75 | 90.42 | 8.43 |

# 4 EXPERIMENTS AND ANALYSIS

## 4.1 EXPERIMENT SETUP

We select a wide range of methods for evaluation and categorize them into two types: detection-only methods, and detection&localization methods.

For detection-only methods, which provide only a binary classification result indicating whether an image is real or synthetic. The methods evaluated in this category are CLIP (Ojha et al., 2023), C2P-CLIP (Tan et al., 2025), DeeCLIP (Keita et al., 2025), DIRE (Wang et al., 2023), FIRE (Chu et al., 2025), Effort (Yan et al., 2024c), DRCT (Chen et al., 2024a), NPR (Tan et al., 2024a), FreqNet (Tan et al., 2024b), AIDE (Yan et al., 2024a), and SAFE (Li et al., 2025a). For DeeCLIP, Effort, DRCT, C2P-CLIP, DIRE, and FIRE, we utilized the pretrained checkpoints from their original papers. For the remaining methods, we use checkpoints open-sourced by (Li et al., 2025b).

We further include three detection&localization methods, which support both detection and localization tasks, to evaluate both their detection and localization performance. These methods are HiFi-Net (Guo et al., 2023), FakeShield (Xu et al., 2024), and SIDA (Huang et al., 2025). For all of these methods, we utilized their publicly released pretrained checkpoints.

For evaluation metrics, we adopt the widely used accuracy (Acc) and average precision (AP). Following (Li et al., 2025b), we also report f.Acc (the accuracy of detecting synthetic images) and r.Acc (the accuracy of detecting real images) to provide a more detailed insight into model performance. For the localization task, we use mean Intersection over Union (mIoU) as the primary metric. Details about our experiment setup can be found in Appendix B.

## 4.2 RESULTS AND ANALYSIS

### 4.2.1 OVERALL RESULTS

We first report the overall results in Table 3. Several key observations can be drawn from the results. First, SAFE outperformed previous methods on the detection task, demonstrating its superiority. Conversely, joint detection&localization methods performed poorly on the detection task. Interestingly, reconstruction-based methods (DIRE, FIRE) perform extremely poorly, indicating their severe bias during training, an observation similar to that in Cazenavette et al. (2024).

Regarding overall generalization, CLIP-based methods (e.g., DeeCLIP) and DRCT exhibit a notable performance drop on the Art split. This indicates that the CLIP backbone may not generalize well to detecting images of different content, which highlights a key shortcoming of several previous

methods and raises concerns about directly applying them to artistic images. In contrast, frequency-based methods demonstrate fine generalization ability on artistic images, suggesting that frequency is indeed a crucial factor in identifying synthetic images, especially when used judiciously (e.g., in AIDE and SAFE).

The specifically trained detection&localization methods show disappointing performance on both tasks. Furthermore, these methods exhibited a significant bias towards making specific types of errors. For example, HiFi-Net and SIDA tended to misclassify synthetic images as real, while FakeShield tended to misclassify real images as synthetic. This suggests a potential robustness problem with these methods.

In summary, our findings highlight a crucial observation: current methods for AI-generated image detection and localization are far from mature and require significant future development. We provide more detailed discussions in Appendix C.

### 4.2.2 DETECTION V.S. LOCALIZATION

Given the existence of detection and localization tasks and the poor performance of detection&localization methods on both, it is natural to ask whether a method's performance trend is similar for both detection and localization tasks, which is **RQ1: Do existing methods perform consistently on detection and localization tasks?**

An observation from Table 3 is that strong detection and localization performance do not necessarily align. For example, SIDA achieves a higher detection accuracy on the Art split compared to the Photo split, yet its localization performance declines on the same split. This seems to suggest that the two tasks can sometimes be at odds with each other.

However, overall detection accuracy is influenced by holistic synthesis models, which may yield noisy results. To delve deeper, we present the f.Acc and mIoU for each partial synthesis model in Figure 3.

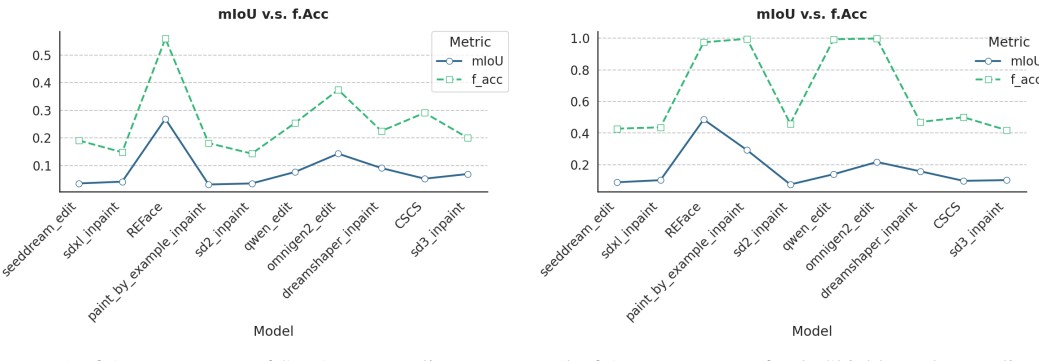

(a) f.Acc v.s. mIoU of SIDA on art split.          (b) f.Acc v.s. mIoU of FakeShield on photo split.

Figure 3: Trends of corresponding f.Acc and mIoU.

A clear observation is that f.Acc and mIoU exhibit nearly identical trends across both methods! This reveals that detection and localization generally do not conflict with each other. Instead, a good localization result generally correlates with a good detection result, highlighting the importance of jointly considering both tasks rather than focusing on only one.

**Takeaway** For detection&localization methods, they perform consistently on detection and localization tasks. Detection and localization do not conflict with each other.

### 4.2.3 GENERALIZATION ACROSS DIFFERENT GENERATIVE MODELS

Given the rapid emergence of new generative models and the varied categories of generative models, it is essential for a detection method to generalize well across different generative models, which is **RQ2: Do existing methods generalize well across different generative methods?**

While we have established that the overall performance of existing models is unsatisfactory—evidenced by their low average accuracy (especially f.Acc) and mIoU—we now investigate the source of this poor performance. Specifically, we aim to determine if it stems from a fundamental lack of overall capability (uniform poor performance across all generative methods) or from a lack of generalization capability (excellent performance on some generative methods but extremely poor performance on others). For detection methods, we present the f.Acc of DRCT and SAFE (two strong detectors as indicated in Table 3) in Figure 4.

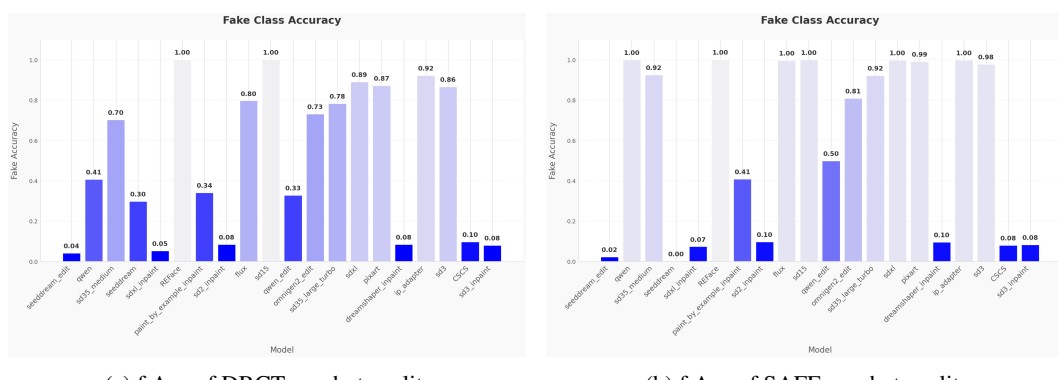

(a) f.Acc of DRCT on photo split.        (b) f.Acc of SAFE on photo split.

Figure 4: f.Acc of different methods. Darker blue indicates smaller values.

Both methods perform poorly due to a lack of generalization capability. They exhibit excellent performance on certain generative models but fail significantly on others. Specifically, they fail on most partial synthesis models while performing well on most holistic synthesis models, indicating that partially generated images represent an overlooked challenge for existing detection methods.

Even for holistic synthesis models, both methods showed almost no effect on SeedDream, a newly released closed-source model. This highlights that, given the rapid development of generative models, generalization capability remains a huge problem for synthetic image detection methods.

We would also like to note that these methods do not completely fail to generalize. For example, both methods perform quite well on REFace (a partial generative method) and Qwen-Image (a recently released and powerful holistic synthesis model), indicating some potential for generalization.

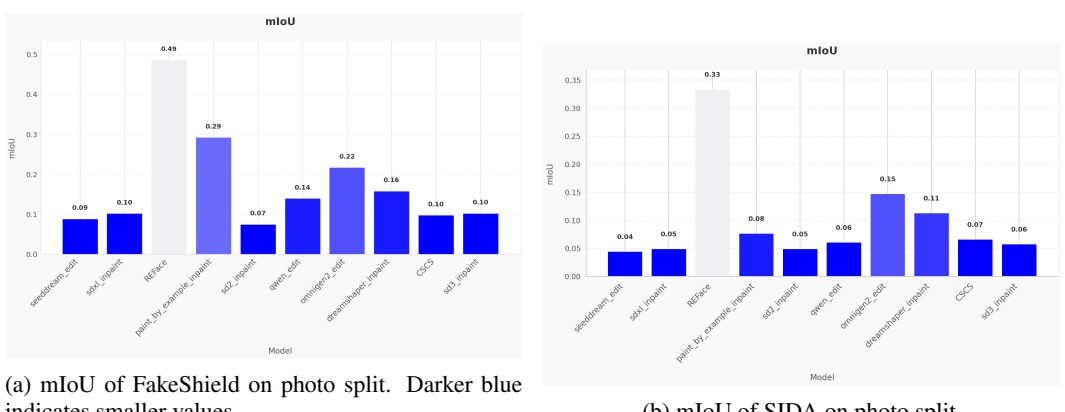

(a) mIoU of FakeShield on photo split. Darker blue indicates smaller values.        (b) mIoU of SIDA on photo split.

Figure 5: mIoU of different methods. Darker blue indicates smaller values.

We then discuss the generalization of localization methods and focus on mIoU. The results are shown in Figure 5, which shows both methods performing poorly on nearly all partial synthesis models. This indicates that the failure of localization methods does not stem from a lack of generalization, but rather from a fundamental lack of overall capability. We present a case study for this circumstance in Appendix C.

**Takeaway** Existing detection-only methods perform poorly on partial synthesis models, highlighting a significant issue with their generalization capability. Conversely, existing detection&localization methods perform poorly across nearly all models tested, suggesting a fundamental lack of overall capability.

### 4.2.4 GENERALIZATION ACROSS DIFFERENT CATEGORIES OF IMAGES

A robust method should perform consistently across different categories of images, including both photographic and artistic images. If it doesn't, its application in real-world scenarios could be problematic. For example, a method proven effective on photos but not on artistic images cannot be used to determine if a drawing was created by a human artist, leading to practical difficulties. This observation leads to our **RQ3: Do existing methods generalize well across different categories of images?**

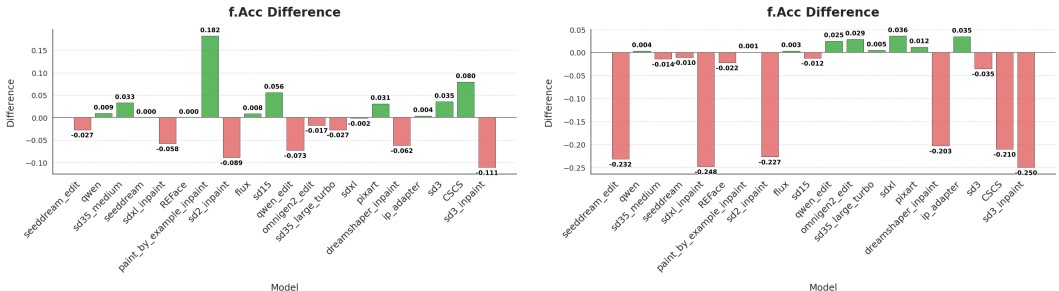

(a) Difference of f.Acc of SAFE between art split and photo split.

(b) Difference of f.Acc of FakeShield between art split and photo split.

Figure 6: The performance difference between art and photo split (art-photo). Red indicates values less than 0 while green indicates values larger than 0.

We calculate the difference in f.Acc for corresponding generative methods between the photo and art splits on both SAFE and FakeShield, and the results are presented in Figure 6. The results show that both methods have a performance difference between the art and photo splits. This difference is generally minimal for SAFE, with most being under 5%. However, FakeShield shows a severe performance degradation on many generative methods in the art split, which indicates a critical failure in image category generalization.

There is an interesting fact that both SAFE and FakeShield show a larger performance difference on partial synthesis models. Specifically, both methods tend to show performance degradation on partial synthesis models in the art split. This indicates that the generalization problem on partial synthesis models is even more severe in the art split, further revealing a generalization problem.

**Takeaway** Existing methods show performance differences between different kinds of image categories, with more notable differences on partial synthesis models, indicating a problem in generalization across contents.

## 5 CONCLUSION

This paper presents UniAIDet, a unified and universal benchmark for AI-generated image content detection and localization. Our benchmark offers comprehensive coverage of diverse image categories and generative model types, and includes a large number of generative models, making it a more robust and useful source for evaluation compared to previous benchmarks with limited scope. We conduct a comprehensive evaluation of a wide range of existing methods using UniAIDet and reveal failures in existing methods. We also answer three key research questions regarding the relation between detection and localization, model generalization, and image category generalization. Overall, our study provides a valuable resource and highlights clear directions for future research.

ETHICS STATEMENT

We adhere to the ICLR Code of Ethics. Our released dataset sources real images from open-source datasets as indicated, following their license and copyright restrictions. Our released dataset containing synthetic images is for research only and does not aim at conveying any information about real-life.

REPRODUCIBILITY STATEMENT

We submit a subset of our dataset through an anonymous GitHub repo to provide an intuitive view of our benchmark. The whole benchmark is too large to be gracefully hosted anonymously, and we will release the whole benchmark upon acceptance. We have disclosed the models and prompts used for generating synthetic images in Section 3.3 and Appendix A. We have also clearly cited the detection and localization methods and discussed the checkpoints used for evaluation in Section 4.1 and Appendix B.

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

## A  BENCHMARK CONSTRUCTION

### A.1  DETAILS ABOUT BENCHMARK CONSTRUCTION

**Details about Mask Generation**    For partial synthesis models, only part of the image is generated; therefore, a mask is needed to identify the generated region for evaluating localization methods. We would like to discuss how this mask is generated in detail. Note that all masks are automatically generated without any human annotation involved.

IMAGE INPAINTING MODELS    As is discussed, given an image $I$, we apply SAM on it to achieve several masks $M = \{m_k\}$, where $m_k = \{(i, j)_k\}$ is a mask. We randomly select a mask $m_{k_0} \in M$, and use it as the mask to perform image inpainting to generate a new image $I'$. We then fuse I and $I'$ following $m_{k_0}$ to achieve the final output synthetic image $\hat{I}$, which satisfies:

$$\hat{I}_{i,j} = \begin{cases} I_{i,j}, & (i,j) \notin m_{k_0} \\ I'_{i,j}, & (i,j) \in m_{k_0} \end{cases} \tag{1}$$

Naturally, the ground truth mask corresponding to $\hat{I}$ is $m_{k_0}$.

IMAGE EDITING MODELS    An image editing models accept an image $I$ and an editing prompt as input and produces an edited image $I'$ as output. Naturally, the mask corresponding to the generated region should be $m_0 = \{(i, j) \mid |I_{i,j} - I'_{i,j}| > 0\}$. However, current image-editing models are not perfect and may modify regions that should not be edited according to the editing prompt. Therefore, to achieve a more reasonable editing output and corresponding mask, we first filter regions with notable modifications, which is $m_1 = \{(i, j) \mid |I_{i,j} - I'_{i,j}| > \tau\}$, where $\tau > 0$ is a hyperparameter.

Note that actually there are multiple channels corresponding to a position $(i, j)$, so we calculate $|I_{i,j} - I_{i,j}'| = \sum_c |I_{i,j,c} - I_{i,j,c}'|$, where $c$ is a corresponding channel.

Further, we filter out too small regions from $m_1$ to reduce noise introduced by image editing methods. Specifically, we first find consecutive regions from $m_1$. Each consecutive region is a non-empty set $m_c \subseteq m_1$ which satisfies $\forall (i,j) \in m_c, (i+1,j) \in m_c \vee (i-1,j) \in m_c \vee (i,j+1) \in m_c \vee (i,j-1) \in m_c$ and $\forall (i,j) \in m_1 - m_c, (i+1,j) \notin m_c \wedge (i-1,j) \notin m_c \wedge (i,j+1) \notin m_c \wedge (i,j-1) \notin m_c$. As can be seen, $m_1 = m_c^{(1)} \cup ... \cup m_c^{(n)}$, and $m_c^{(i)} \cap m_c^{(j)} = \phi$. We construct the final mask $m = \bigcup_{i,|m_c^{(i)}|>\gamma} m_c^{(i)}$, where $\gamma > 0$ is a hyperparameter. In this study, we select $\tau = 40$ and $\gamma = 20$.

After achieving this final mask, we apply a similar operation as above by fusing $I$ and $I'$ following $m$, which is:

$$\hat{I}_{i,j} = \begin{cases} I_{i,j}, & (i,j) \notin m \\ I_{i,j}', & (i,j) \in m \end{cases} \tag{2}$$

DEEPFAKE METHODS    Each deepfake method takes an image $I$ and a face image $I_f$ as input and produces an image $I'$ as an output. Fortunately, each deepfake method already produces a mask $m$ indicating the modified region, so there is no need to specifically find the mask. We perform a similar fusing operation to achieve the final synthetic image $\hat{I}$:

$$\hat{I}_{i,j} = \begin{cases} I_{i,j}, & (i,j) \notin m \\ I_{i,j}', & (i,j) \in m \end{cases} \tag{3}$$

**Details about Quality Check**    For text-to-image, image-to-image, and image inpainting methods, there is no need to verify the quality of the generated image, as the goal of our benchmark is to faithfully reflect the performance of existing image generative models. Manual selection may introduce additional bias.

For image editing and deepfake methods, we sampled several generated images to verify whether they faithfully follow the instruction (editing instruction or face swapping). Our selected models generally follow instructions well, therefore we include them to construct our test set.

Regarding the mask generation for image editing methods, we select all large consecutive regions as the edited part without referring to the editing instruction. Our rationale is to best handle the diversity of editing instruction, including foreground, background, single-object, multiple-object, style, etc. We believe this design choice best reflects the real performance of current image editing models.

Also, for closed-source models like SeedDream and SeedDream-Edit, since they have too strong filters, many generated results are marked as unsafe and not responded to, resulting in a relatively small amount of data belonging to these models. However, the number of images belonging to these models is still reasonable as indicated in Figure 1.

A.2    DETAILS ABOUT PROMPT GENERATION

An open-source MLLM (Gemma3-4B) is used to generate image captions (for text-to-image and image-to-image generation) and image editing instructions (for image editing generation). We provide prompt details as follows:

We manually sampled and checked the generated captions and editing instructions and verified that they are high-quality. We present examples as follows.

A.3    INSIGHTS BEHIND OUR BENCHMARK CONSTRUCTION

Just like we have stated in Section 3.1, photo-realism is not a key factor we focus on when constructing our benchmark, and we did not specifically choose samples that appear more indistinguishable

Image Captioning Prompt

You are a powerful MLLM caption generator.
Your task is to describe the content of the given image in detail.
You should provide a comprehensive description of the image, including objects, scenery, actions, and any other relevant details.
You should also include style, color, lighting if applicable.
Do not make any assumptions about the image.
If you are unsure about certain aspects of the image, describe only what you can clearly see.
Your response should only contain the description without any additional commentary or questions.

Figure 7: Image captioning prompt.

Editing Instruction Generation Prompt

You are a powerful MLLM image editing prompt generator.
Your task is to provide a possible editing instruction based on the given image.
You should provide a reasonable editing instruction which prompts modifying the content of the image, such as add, remove or modify some content or style in the image.
Your response should only contain the instruction without any additional commentary or questions.

Figure 8: Image editing instruction generation prompt.

Image Captioning Example

A cartoon-style illustration of a young female character with long, flowing red hair. The hair is styled with a large, swirling curl on the right side of her head and cascades down her back. Her eyes are a dark red color and she has a slight, downward gaze. She is wearing a gray jacket over a dark shirt with a visible, patterned collar. A small, ornate pendant hangs around her neck, featuring a gold-colored bell-shaped element and a small, metallic-looking object. The background is solid white.

Figure 9: Image captioning example.

Image Editing Instruction Example

Add a sparkling magical effect to the staff.

Figure 10: Image editing instruction.

to humans because we believe that this may introduce additional bias to our constructed dataset. Different methods identify and localize synthetic images in different ways, and our goal is to provide a benchmark faithfully representing the natural distribution of synthetic images to support fair comparison between different methods.

## B  EXPERIMENT SETUP

### B.1  METRIC CALCULATION

We view the detection task as a binary classification task, which yields four statistics: True Positive Number (TP), True Negative Number (TN), False Positive Number (FP), and False Negative Number (FN). The metrics are calculated as:

$$Acc = \frac{TP + TN}{TP + TN + FP + TN} \tag{4}$$

$$f.Acc = \frac{TP}{TP + FN} \tag{5}$$

$$r.Acc = \frac{TN}{TN + FP} \tag{6}$$

For mIoU, consider a ground truth mask as $G = \{(i, j)\}$ and a predicted mask as $G^{'} = \{(p, q)\}$, the mIoU is the average of IoU, which is the average of $\frac{|G \cap G^{'}|}{|G \cup G^{'}|}$. We only calculate mIoU on partial synthesis images.

### B.2  CHECKPOINT USAGE

As discussed, we did not train any method. Instead, for DeeCLIP, Effort, DRCT, C2P-CLIP, DIRE, and FIRE, we utilized the pretrained checkpoints from their original paper, and for the remaining methods, we used checkpoints open-sourced by (Li et al., 2025b) for convenience. Regarding detection&localization methods, we use SIDA-13B, FakeShield-22B, and the open-source HiFi-Net checkpoints.

It is worth noting that these checkpoints may bear slightly different training protocols and datasets due to the differences in their design and implementation. However, this difference is beyond our discussion, and the goal of our evaluation is to reveal method performance under a reasonable setup. If a method is trained using generated images from all models used to construct our benchmark, its performance will be almost perfect, yet this is not a reasonable scenario. Our evaluation follows a reasonable and practical evaluation protocol, representing real-world scenarios well.

## C  ADDITIONAL ANALYSIS

### C.1  DETAILED ANALYSIS OF METHODS

We present some additional discussions about the methods evaluated. First of all, we would like to again note that we did not conduct any training on any specific method. Instead, we use the open-source trained checkpoints for evaluation. Therefore, if there are some issues during training (e.g. DIRE (Wang et al., 2023) as suggested by (Cazenavette et al., 2024)), the evaluated result may differ a lot from the reported values in their original paper. However, we argue that this is exactly the value of our proposed benchmark - to provide a comprehensive and faithful evaluation of existing methods.

Apart from DIRE (Wang et al., 2023) and FIRE (Chu et al., 2025), most detection-only methods show superior performance on identifying real images. This observation suggests that identifying synthetic images is still a giant problem. In contrast, detection&localization methods are bad at both real images and synthetic images, indicating a problem in both capabilities.

## C.2  CASE STUDY: EDITING SCENARIOS

We present several typical cases to offer a qualitative understanding of existing methods. Specifically, we use SIDA as an example to illustrate the behavior of current detection&localization models. Our analysis focuses on the end-to-end image editing scenario, a context that has been generally overlooked by previous benchmarks.

Table 4: Examples of localization results of SIDA on SeedDream Edit.

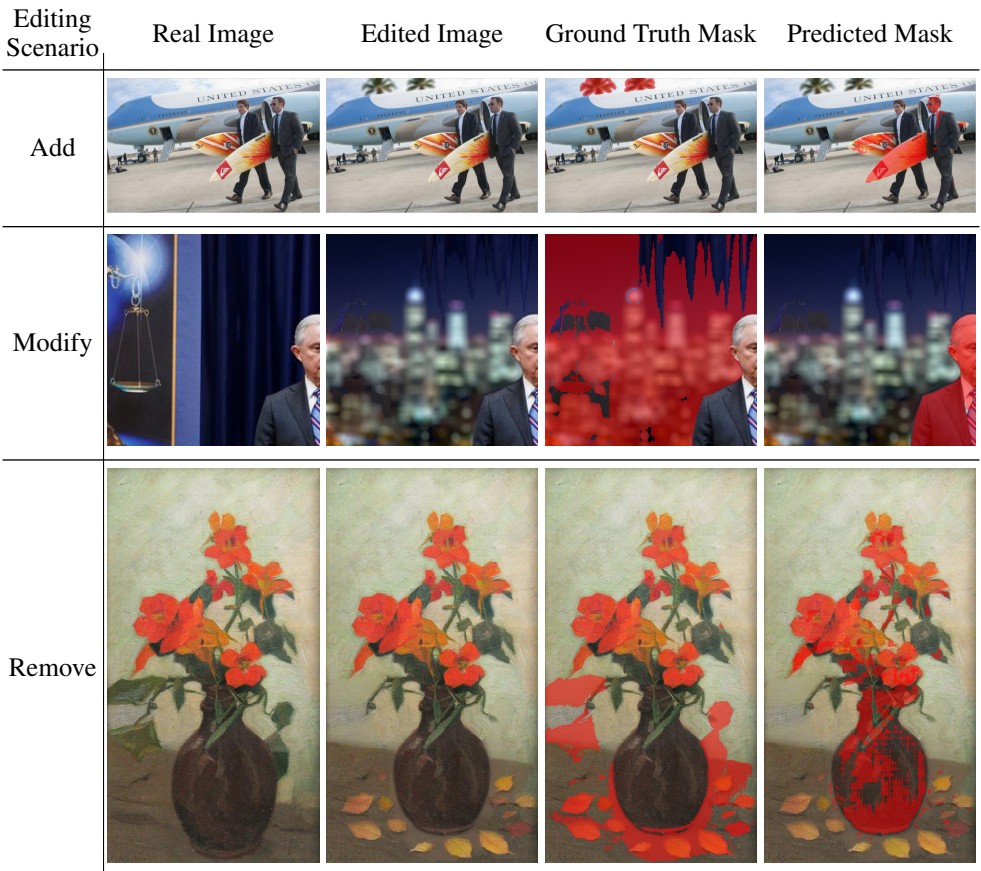

| Editing Scenario | Real Image | Edited Image | Ground Truth Mask | Predicted Mask |
|---|---|---|---|---|
| Add | | | | |
| Modify | | | | |
| Remove | | | | |

We selected different scenarios and image categories, and the results reveal an interesting trend: current localization methods tend to predict "foreground distinguishable objects.", even if the ground truth is sometimes background or not distinguishable objects. This is likely because they are fine-tuned on segmentation models, which are designed for localizing foreground distinguishable objects. As a result, they predict surfboards in the "add" scenario, the person in the "modify" scenario, and the vase in the "remove" scenario. However, editing can occur anywhere, which leads to this biased performance, as shown in Table 4. Therefore, we argue that end-to-end image editing with arbitrary editing prompts provides a practical and challenging scenario for AI-generated image detection&localization methods, which should be paid more attention to.

## D  USE OF LLMS

We use LLM to assist writing, including polishing expressions and correcting grammar and word usage. LLM is also used for finding related works. All LLM-generated content is verified by humans before they are included in the manuscript.

