# OpenReview forum: "UniAIDet: A Unified and Universal Benchmark for AI-Generated Image Content Detection and Localization"
_ICLR.cc/2026/Conference — ICLR 2026 Conference Withdrawn Submission_

### Official Review · Reviewer_uwTV · 2025-10-27

**Soundness:** 2
**Presentation:** 2
**Contribution:** 2
**Rating:** 2
**Confidence:** 5

**Summary:**

This paper introduces UniAIDet, a comprehensive benchmark dataset for AI-generated image detection and localization. UniAIDet covers diverse image generation methods such as text-to-image, image inpainting, and  DeepFake, and a wide range of generative models, including both open-source and closed-source ones. By testing existing AI-generated image detection and localization methods on UniAIDet, it is concluded that: (1) their generalization performance is still unsatisfactory; (2) their performance on detection and localization is generally consistent; (3) detection-only methods have poor generalization to partially synthesized images, while multi-task detection and localization methods perform poorly on all types of generated images.

**Strengths:**

1. The proposed dataset is diverse and comprehensive. It covers various generative methods (both holistic and partial synthesis), a broad range of generative models, and both realistic and artistic images.
2. The benchmarking of existing detection and localization methods reveals some interesting phenomena, for example, (1) some methods (e.g., NPR, AIDE) have consistent performance of detecting realistic and artistic images, even though they are trained solely on realistic ones; (2) models trained for fully generated image detection struggle to detect partially generated images.

**Weaknesses:**

1. A major limitation of Sec. 4 is the **lack of in-depth analysis** of the evaluation results and observations. Limited insights are provided regarding the reasons behind the phenomena and possible future directions for improving the detection and localization methods.
2. The claim that the proposed UniAIDet is "the **first** large-scale, wide-coverage benchmark AI-generated image content detection and localization" (Lines 96-97) needs to be more specific about the coverage and uniqueness (e.g., the inclusion of both realistic and artistic images, and the range of generative models). Otherwise, this could seem like an overclaim.
3. The contributions listed in Lines 96-103 are too general. It would be better to clarify what **new observations** are made based on the proposed dataset, especially those that cannot be revealed by existing datasets.
4. The conclusion that methods based on the CLIP backbone may not generalize well to artistic images (Lines 322-324) is not sufficiently supported by the results in Table 3. The reason for their inferior performance on artistic images may also be due to the training data and methods, rather than the CLIP backbone. (Suggestion: consider testing the DRCT pre-trained models with ConvB and CLIP backbones for a more rigorous comparison.)
5. The conclusion that the detection and localization tasks do not conflict with each other (Lines 371-372) is only evidenced in Figure 3, where only the performance of two selected models is shown, and the "consistency" is more intuitive and qualitative rather than supported by statistical analysis.

**Questions:**

1. Are the evaluation results in Table 3 possibly affected by the compression bias [1]? Specifically, if the real images are stored with lossy compression formats like JPEG, while the generated images are free from lossy compression, the performance of some generators that learned such compression bias from their training data can be overestimated.

[1] Fake or JPEG? Revealing Common Biases in Generated Image Detection Datasets. ECCV 2024 Workshop.

---

### Official Review · Reviewer_XW4i · 2025-10-27

**Soundness:** 3
**Presentation:** 2
**Contribution:** 3
**Rating:** 6
**Confidence:** 2

**Summary:**

This paper proposes a unified, large-scale benchmark, UniAIDet, aiming to evaluate AI-generated image detection and localization tasks. The core contribution of the paper lies in its systematic integration: covering 20 generative models, 80k images, and two major scenarios: photography and art.

**Strengths:**

1. It mainly covers full synthesis and partial synthesis. It also includes photo, art, and localization tasks, making it very comprehensive.

2. The authors provide clear mathematical definitions for detection vs. localization: centering on the criterion of "whether pixels are produced by a generative model" rather than artifact-based methods. This makes the research direction more rigorous.

**Weaknesses:**

1. The paper focuses more on empirical comparisons and lacks mechanistic explanations. Why is partial synthesis particularly difficult to detect? The feature differences across different model types (frequency/spatial/semantic) are not visualized in depth.

2. It primarily considers generalization detection. What about robustness? Attacks like cropping and then re-generation / secondary editing (regeneration attacks).

**Questions:**

1. Why is partial synthesis particularly difficult to detect?

2. The relationship between feature differences across different model types (frequency/spatial/semantic) and the difficulty of detection.

3. How do these detectors change when facing certain attacks?

---

### Official Review · Reviewer_FHFf · 2025-10-31

**Soundness:** 2
**Presentation:** 3
**Contribution:** 2
**Rating:** 2
**Confidence:** 5

**Summary:**

This paper introduces a unified benchmark called UniAIDet, designed to jointly evaluate both detection of AI-generated content and pixel-level localization of manipulated regions, covering both fully generated images and partially edited / face-swapped images. The dataset is reported to contain approximately 80,000 real and generated images spanning around 20 generation or editing models, and to include multiple domains such as photographic content and artistic/anime-style content. Using this benchmark, the authors evaluate existing AIGC detection methods and observe that current approaches still exhibit notable weaknesses.

**Strengths:**

- The paper is clearly written. The experiments are presented in an organized and systematic way, with diverse forms of visualization, including a large number of statistical figures.
- The authors conduct a unified evaluation of multiple existing detectors, and the experimental results are comprehensive.

**Weaknesses:**

- The paper claims UniAIDet as “the first large-scale, wide-coverage benchmark … covering most potential practical scenarios,” emphasizing breadth across both holistic and partial synthesis, plus localization. However, [1] similarly targets localization, cross-domain generalization, and includes explicit explanatory annotations. A more direct comparison with [1] is needed.
- The dataset construction process involves no human validation. Although an NSFW detector is applied to filter real images, there remains a potential safety/abuse concern for AIGC images.
- The paper currently only releases an anonymized subset, and the provided link appears to be inactive. Given that the full dataset is not publicly available and that the benchmark may include copyrighted material and face-swapped content with potential privacy implications, it is unclear whether the community can safely reproduce and use this benchmark. The current ETHICS STATEMENT feels too lightweight in this regard.
- For the instruction-guided editing setting, regions are obtained via pixel-difference thresholding between the original and edited images. Is this region truly equivalent to “the area modified according to the instruction”? Was there any manual verification of this consistency?

[1] So-Fake: Benchmarking and Explaining Social Media Image Forgery Detection

**Questions:**

- In Table 2, there are two typos in the method names: "AIGCDetct" should be corrected to "AIGCDetect", and "Chalmeon" should be corrected to "Chameleon".
- The title of Section 2 should be revised from "RELATED WORKS" to "RELATED WORK", which is the conventional form in academic writing.
- In Equation (4), the denominator contains two repeated TN terms. Please check and correct the expression to avoid duplication.

---

### Note · Authors · 2026-01-22

I have read and agree with the venue's withdrawal policy on behalf of myself and my co-authors.